# Stimulus relevance modulates contrast adaptation in visual cortex

Andreas J Keller[1,2,3]\*, Rachael Houlton[4], Björn M Kampa[2,5,6], Nicholas A Lesica[7], Thomas D Mrsic-Flogel[4,8], Georg B Keller[3,9†], Fritjof Helmchen[2†]

[1]Institute of Neuroinformatics, University of Zurich and ETH Zurich, Zürich, Switzerland; [2]Brain Research Institute, University of Zurich, Zürich, Switzerland; [3]Friedrich Miescher Institute for Biomedical Research, Basel, Switzerland; [4]Department of Neuroscience, Physiology and Pharmacology, University College London, London, United Kingdom; [5]Department of Neurophysiology, Institute of Biology II, RWTH Aachen University, Aachen, Germany; [6]JARA BRAIN Institute for Neuroscience and Medicine, Forschungszentrum Jülich, Jülich, Germany; [7]Ear Institute, University College London, London, United Kingdom; [8]Biozentrum, University of Basel, Basel, Switzerland; [9]Faculty of Natural Sciences, University of Basel, Basel, Switzerland

\*For correspondence: andi@ini.ethz.ch

†These authors contributed equally to this work

**Abstract** A general principle of sensory processing is that neurons adapt to sustained stimuli by reducing their response over time. Most of our knowledge on adaptation in single cells is based on experiments in anesthetized animals. How responses adapt in awake animals, when stimuli may be behaviorally relevant or not, remains unclear. Here we show that contrast adaptation in mouse primary visual cortex depends on the behavioral relevance of the stimulus. Cells that adapted to contrast under anesthesia maintained or even increased their activity in awake naïve mice. When engaged in a visually guided task, contrast adaptation re-occurred for stimuli that were irrelevant for solving the task. However, contrast adaptation was reversed when stimuli acquired behavioral relevance. Regulation of cortical adaptation by task demand may allow dynamic control of sensory-evoked signal flow in the neocortex.

## Introduction

Our sensory systems constantly receive streams of sensory signals. The computational resources to process this input, however, are limited. Neural circuits in sensory systems have been shown to reduce responses to sustained stimuli (*Adrian and Zotterman, 1926*; *Albrecht et al., 1984*; *Maffei et al., 1973*) or selectively enhance aspects of the sensory input that are relevant to a behavioral task (*Desimone and Duncan, 1995*; *Ito and Gilbert, 1999*; *Kato et al., 2015*; *Reynolds and Heeger, 2009*; *Zhang et al., 2014*). In visual cortex, neural responses to a sustained stimulus adapt over the course of a few seconds. Thus, a proposed function of adaptation is to redistribute processing resources to behaviorally relevant or novel stimuli. Most experiments on adaptation, however, were carried out in anesthetized animals. While sensory-evoked responses are known to be modulated by task engagement or attention (*Ito and Gilbert, 1999*; *Harris and Mrsic-Flogel, 2013*; *Reynolds and Chelazzi, 2004*), it is still unclear if cortical adaptation is modulated by the behavioral relevance of the stimulus.

## Results and discussion

To test the behavioral dependence of cortical response adaptation, we presented sustained moving grating stimuli to mice in different behavioral states and in conditions with different behavioral relevance of the visual test stimulus. Consistent with previous findings (*Ahmed et al., 1997*; *Carandini and Ferster, 1997*; *Sanchez-Vives et al., 2000*; *Vidyasagar, 1990*; *Keller and Martin, 2015*), we found that the responses of neurons in anaesthetized mouse primary visual cortex (V1) adapt to sustained high-contrast grating stimuli (*Figure 1*), and that this adaptation depends on local cortical activity (*King et al., 2016*) (*Figure 1—figure supplement 1*). Several mechanisms have been proposed to underlie such contrast adaptation (*Ahmed et al., 1997*; *Carandini and Ferster, 1997*; *Sanchez-Vives et al., 2000*; *Vidyasagar, 1990*; *Keller and Martin, 2015*), including tonic feedforward inhibition mediated by parvalbumin positive (PV+) interneurons (*Ahmed et al., 1997*; *Keller and Martin, 2015*). Accordingly, we found that PV+ neurons adapt less than putative excitatory neurons and that adaptation is only weakly orientation-specific for both neuron types (*Figure 1—figure supplement 2*). To test if neural responses also adapt in awake mice, we compared adaptation measured in the same neurons using two-photon calcium imaging under anesthesia and during wakefulness. As opposed to data obtained under anesthesia, we found that adaptation was absent and neural activity even increased during sustained grating presentations in awake recordings (*Figure 1* and *Figure 1—figure supplements 2i–l*, *3* and *4*). Adaptation was stronger (i.e. slope of adaptation more negative) for almost all cells in anesthetized compared to awake mice (*Figure 1—figure supplement 3a*). This reversal of adaptation in awake mice could be explained neither by response saturation, nor locomotion, nor eye movements (*Figure 1d* and *Figure 1—figure supplements 2i,k* and *3*). We hypothesize that an attentional mechanism could prevent adaptation when a stimulus is of unknown relevance to the animal. If so, adaptation should reappear if mice divert attention away from the stimulus and learn that the stimulus is behaviorally irrelevant.

To test the role of stimulus relevance for adaptation, we designed a simple visual navigation task (*Figure 2*), in which mice were trained to run to reach the end of a virtual tunnel using visual feedback, while a drifting grating was presented in a fixed part of the visual field (probe patch, centered on the retinotopic location of the recording site; see Materials and methods). Consistent with the lack of adaptation in the passively observing awake mouse (*Figure 1*), we found that adaptation to the grating stimulus was absent initially. As mice learned to perform the navigation task, however, adaptation of neural responses to the grating stimulus reappeared ('grating-irrelevant' condition, *Figure 2e,g,h*, *Figure 2—figure supplement 1a*, *Video 1*). This reappearance of adaptation suggests that mice, as they learned to interact with the task-relevant part of the visual field, diverted attention away from the grating stimulus that contained no task-relevant information. Based on this finding, we predicted that for an identical visual input, but when the grating stimulus is behaviorally relevant, neural responses should not adapt with experience. To test this prediction, we showed a different group of mice a replay of the visual stimulus sequence generated by one of the mice in the grating-irrelevant group but increased the behavioral relevance of the grating stimuli by delivering a water reward at the offset of the grating ('grating-relevant' condition, *Figure 2f–h*, *Figure 2—figure supplement 1b*). We found that adaptation remained absent over training sessions in the grating-relevant group, despite visual experience being identical to the grating-irrelevant group. Moreover, when mice exhibited anticipatory licking to the reward, neural responses showed an effect opposite to adaptation and activity increased over the course of the stimulus presentation (*Figure 2—figure supplement 2a–i*). We verified that the differences in adaptation between the grating-relevant group and the grating-irrelevant group cannot be explained by learning-related changes in mean running speed, time spent running or number of saccades (*Figure 2—figure supplement 2j–l*). These results suggest, therefore, that in behaving animals contrast adaptation is modulated bidirectionally by stimulus relevance (*Figure 2—figure supplement 2f,i*).

Thus, the responses of neurons in layer 2/3 of V1 do not adapt to sustained stimuli that are behaviorally relevant, but they do adapt if the stimulus within their receptive field is irrelevant and animals learn to direct attention away from it to other parts of the visual field. These effects are likely mediated by attentional mechanisms (*Zhang et al., 2014*; *Kim et al., 2016*) that could directly enhance the responses to relevant stimuli to prevent adaptation. The attentional modulation of

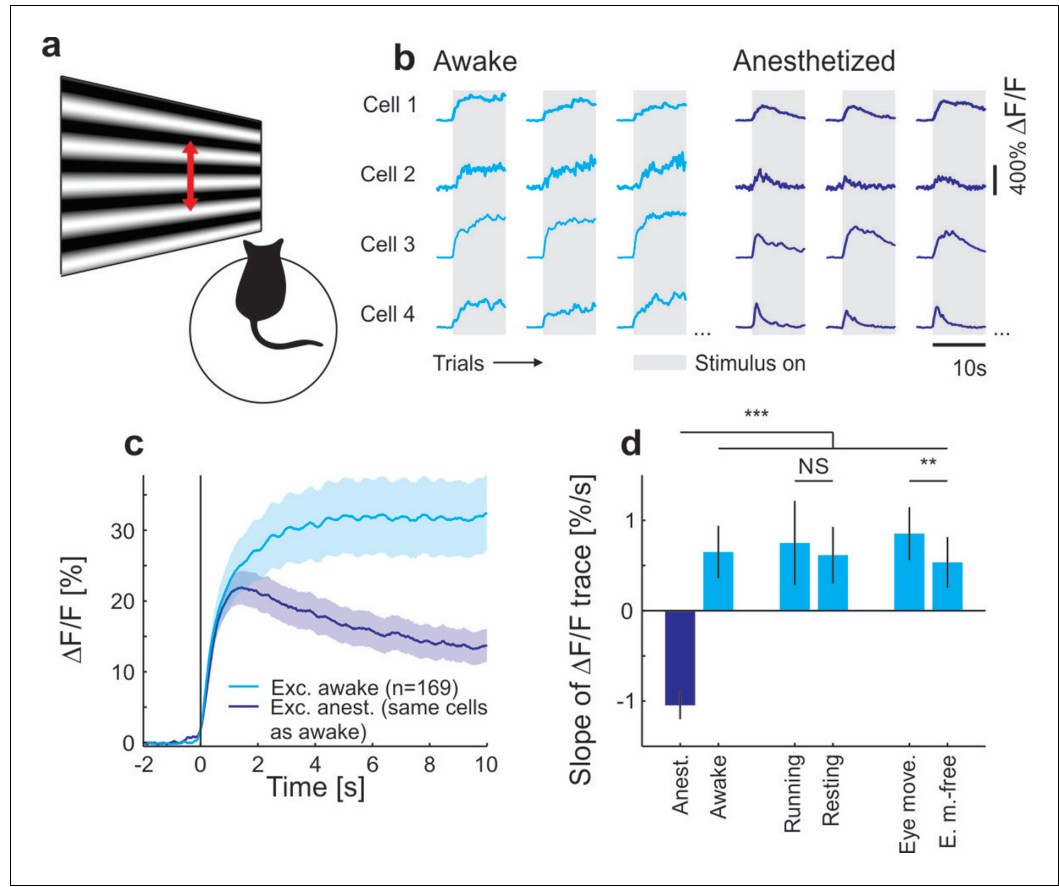

**Figure 1.** Contrast adaptation in awake and anesthetized mice. (**a**) Schematic of the experimental setup. Calcium imaging with GCaMP6m (*Chen et al., 2013*) was performed during the presentation of drifting sinusoidal gratings. (**b**) Calcium transients from four example putative excitatory cells tuned to a moving sinusoidal grating at 50% contrast (presented for 10 s; grey shadings). The same cells were recorded during wakefulness and anesthesia. (**c**) Averaged calcium responses of tuned putative excitatory cells. Note that even small differences in adaptation can be detected using two-photon imaging (*Figure 1—figure supplement 2a–d*). Curves plotted as mean ± SEM (shading). (**d**) Slope of adaptation of single cells recorded in different behavioral states (same data as in **c**; line fit to the data in time window 1–9.75 s). Anesthetized mice show a significantly more negative slope compared to all other states [anest. (169 cells) – awake (169 cells): $p<10^{-10}$; Wilcoxon signed-rank; running (51 cells): $p<10^{-4}$; resting (169 cells): $p<10^{-10}$; eye movements (168 cells): $p<10^{-10}$; eye movement-free (165 cells): $p<10^{-10}$; Wilcoxon rank-sum]. There was no significant difference between running and resting mice, as opposed to the significant but small difference in eye movement and eye movement-free trials (p=0.49 and p=0.0047, respectively; Wilcoxon rank-sum). NS, not significant; **p<0.005; ***p<0.0005.

The following figure supplements are available for figure 1:

**Figure supplement 1.** Adaptation in visual cortex of anesthetized mice is prevented by optogenetic silencing of cortical neurons (see also *King et al. 2016*).

**Figure supplement 2.** Differences in contrast adaptation across cell-types can be revealed using two-photon imaging.

**Figure supplement 3.** Contrast adaptation in anesthetized compared to awake mice and effects of running and eye movements on adaptation in awake mice.

**Figure supplement 4.** A large fraction of neurons in awake mice were suppressed during the stimulus and decreased their activity below baseline.

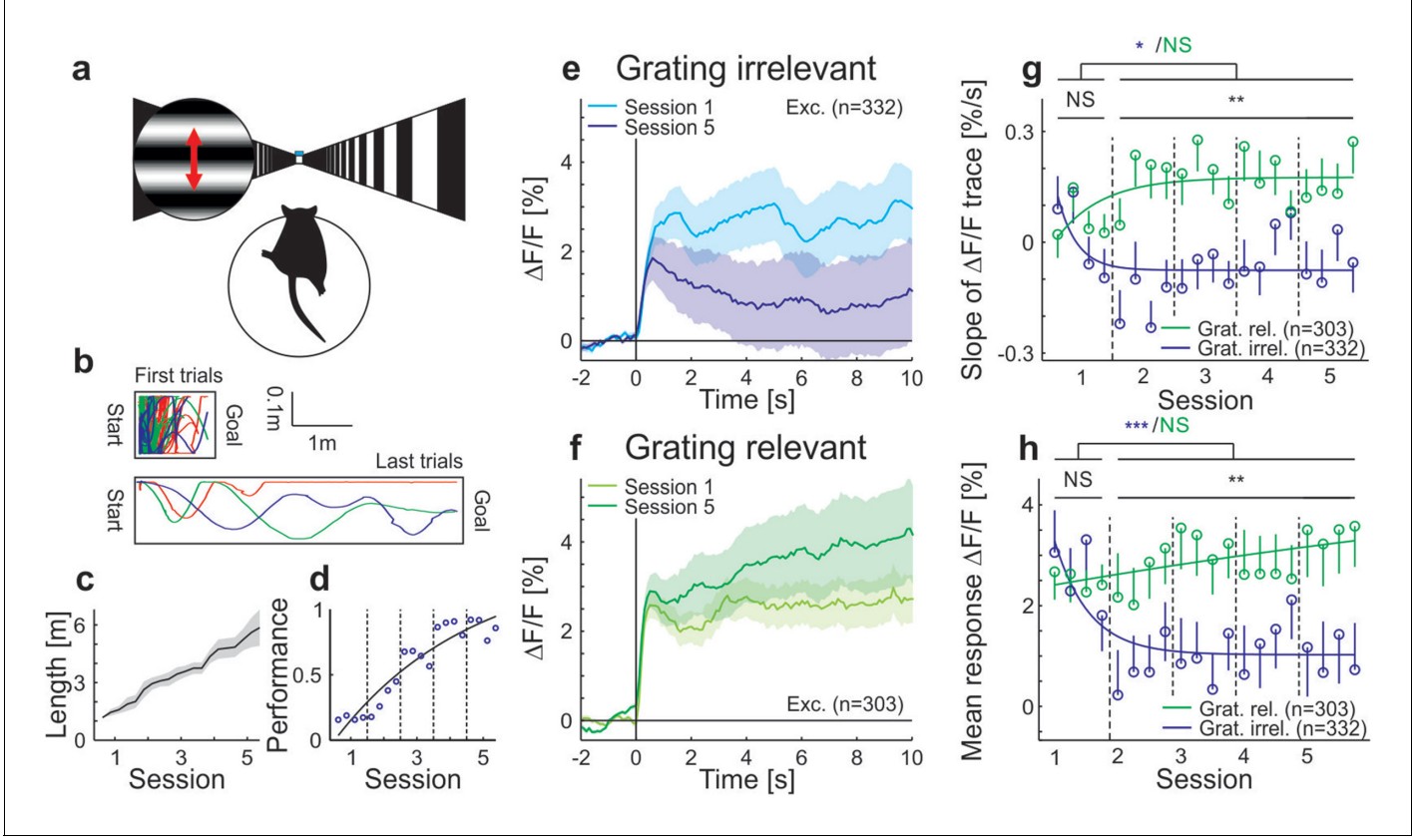

**Figure 2.** Adaptation is modulated by stimulus relevance in awake mice. (**a**) Schematic of the behavioral task. For the grating-irrelevant condition, movement of a virtual tunnel projected on a toroidal screen was coupled to the locomotion (rotation and running on a spherical treadmill) of the head-restrained mice (**Dombeck et al., 2007**). Mice were trained to orient and run to the end of the tunnel for a water reward. We presented a horizontal sinusoidal moving grating in a circular probe patch centered on the retinotopic location of the recording site, interspersed with random intervals of gray (10–20 s; **Video 1**). (**b**) First and last paths of a sample mouse (four days apart). The colors show individual trials. (**c**) Task difficulty (length of the tunnel) was increased over learning to keep the number of rewards approximately constant. (**d**) Learning curve of an example mouse (solid line: exponential fit). The performance is quantified as the fraction of time spent running in the direction of the goal (±25°). (**e**) Data from animals trained in the behavioral task under grating-irrelevant conditions. Traces show averaged calcium responses (GCaMP6f) (**Chen et al., 2013**) of tuned putative excitatory cells to a moving sinusoidal grating. Curves plotted as mean ± SEM (shading). (**f**) Same as **e** but for animals exposed to the grating-relevant condition. For this condition, the visual stimulus on the screen was a replay of the visual flow from one of the mice in the grating-irrelevant group. To match the initial responses of the grating-relevant and the grating-irrelevant traces, ten percent of neurons were excluded from analysis (see Materials and methods). Note that this did not change the results. (**g**) Slopes of adaptation of the same cells as in **e** and **f** (line fit to the data in time window 1–10 s). In the grating-irrelevant condition, the slope significantly decreases from the first to the following sessions, as opposed to the grating-relevant condition (putative excitatory: 332 and 303 cells, respectively; p=0.017 and p=0.28, respectively; Wilcoxon signed-rank). The slopes for the two conditions are similar during the first session, but significantly differ during later sessions (putative excitatory: 332 and 303 cells; p=0.84 and p=0.0036, respectively; Wilcoxon rank-sum). The solid curves are exponential fits to the data. Error bars represent mean ± SEM. (**h**) Same as **g** but for mean response to the grating. In the grating-irrelevant condition, the mean response significantly decreases from the first to the following sessions, as opposed to the grating-relevant condition (putative excitatory: 332 and 303 cells, respectively; p<10$^{-4}$ and p=0.85, respectively; Wilcoxon signed-rank). The mean responses for the two conditions are similar during the first session, but significantly differ during later sessions (putative excitatory: 332 and 303 cells; p=0.61 and p=0.0015, respectively; Wilcoxon rank-sum). NS, not significant; *p<0.05; **p<0.005; ***p<0.0005.

The following figure supplements are available for figure 2:

**Figure supplement 1.** Scatterplots showing slopes of adaptation of the cells in training session one compared to the average slope in sessions 2–5 in awake mice (see also **Figure 2**; line fit to the data in time window 1–10 s).

**Figure supplement 2.** Licking, running, and eye-movement behavior in awake mice.

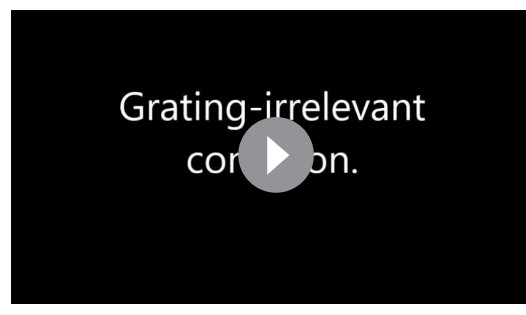

**Video 1.** Sample mouse under grating-irrelevant condition in session 4.

adaptation was also not simply explained by changes in the adaptation of inhibitory neurons (data not shown) and is unlikely to be generated only locally.

In contrast to our findings in mouse V1, fMRI studies on awake humans have found response adaptation in V1 upon the presentation of visual patterns (*Gardner et al., 2005*; *Fang et al., 2005*; *Huk and Heeger, 2001*, but see *Kastner et al., 2004*). This disparity is likely the result of small but relevant differences in study design. For example, *Huk and Heeger (2001)* find a weak adaptation in V1 when participants were attending to two separate moving plaid stimuli. This could be explained by the fact that distributing attention decreases attentional effects (*Ito and Gilbert, 1999*). Other studies (*Gardner et al., 2005*; *Fang et al., 2005*) used stimuli that are known to cause adaptation in thalamus and even the retina (*Smirnakis et al., 1997*; *Chander and Chichilnisky, 2001*) and cannot distinguish the effects of cortical adaptation from those of subcortical adaptation.

In summary, we have shown that adaptation is dynamically regulated by task demand during learning. Our data are consistent with the idea that cortex dynamically regulates the flow of sensory information by suppressing responses to non-relevant stimuli through mechanisms of adaptation, while boosting sensory responses that are behaviorally important.

## Materials and methods

### Animals

All experiments and surgical procedures were carried out in accordance with the UK Animal (Scientific Procedures) Act under project license 70/7573, approved by the Cantonal Veterinary Office of Zurich, Switzerland, under license number 62/2011, or by the Cantonal Veterinary Office of Basel-Stadt, Switzerland, under license number 2537.

For the electrophysiology experiments, we used transgenic mice selectively expressing channelrhodopsin-2 in parvalbumin-positive neurons (PV+). These mice were generated by crossing *Pvalb^Cre* (Jackson 008069) and *Ai32* animals (Jackson 012569). Data were collected from six mice (two female, four male, P39-P83).

For the two-photon experiments on mice not engaged in visually-guided behavior, we used transgenic mice selectively expressing tdTomato in PV+ neurons. These mice were generated by crossing *Pvalb^Cre* mice (Jackson 008069) with the *Ai14* reporter line (Jackson 007914). Data were collected from eight adult mice (two female, six male, P90-161).

For the two-photon experiments on mice engaged in visually-guided behavior (grating-irrelevant and grating-relevant condition), we used transgenic mice selectively expressing tdTomato in GABAergic neurons. These mice were generated by crossing *Slc32a1^Cre* mice (Jackson 016962) with the *Ai9* reporter line (Jackson 007909). Data were collected from seven mice (two female, five male, P80-P282) for the grating-irrelevant condition and six mice (one female, five male, P80-P286) for the grating-relevant condition.

### Surgical procedures and anesthesia

For the electrophysiology experiments, animals were anesthetized with a mixture of Fentanyl (Sublimaze, 0.05 µg/g of body weight), Midazolam (Dormicum, 5.0 µg/g of body weight) and Medetomidine (Domitor, 0.5 µg/g of body weight) injected intraperitoneal (i.p.). An adequate depth of anesthesia was indicated by lack of response to toe pinch. Eye cream (Isoptomax) was applied to the eyes to prevent dehydration during surgery. Atropine Sulphate (Hameln Pharmaceuticals, 0.02 µg/g of body weight) and Dexamethasone Sodium Phosphate (Hospira, 0.8 µg/g of body weight) were injected subcutaneously to reduce secretions and edema, respectively. Cortex buffer solution (125 mM NaCl, 5 mM KCl, 10 mM glucose, 10 mM HEPES, 2 mM MgSO$_4$, and 2 mM CaCl$_2$ [pH 7.4], 50

µl) was also injected subcutaneously to prevent dehydration. Throughout the experiment, the body temperature was maintained at 38°C, measured with a rectal probe and controlled with a heating blanket. Fur was trimmed and an incision was made at the rear of the head, approximately level with the ears. The skull was cleared of tissue and immobilized by affixing it to a metal head plate using dental cement (Paladur, Heraeus Kulzer). The plate was then secured in a frame with the head in a horizontal position. A small craniotomy (~2 mm diameter) was made above the right monocular primary visual cortex (V1), determined by stereotaxic coordinates, using a high-speed foot-operated drill (Foredom). The exposed cortical surface was kept moist with cortex buffer solution (see above). The dura was removed and the cortex was covered with 2% agarose following initial electrode array insertion. After surgery, the eye cream was removed except for a thin layer, keeping the eye moist whilst minimizing any visual disturbance.

For the two-photon experiments on mice not engaged in visually-guided behavior, the implantation of the hook for head fixation and the virus injection were performed in two separate surgeries. First, the animals were anesthetized with isoflurane (1–2%). Throughout the surgery, body temperature was measured and controlled with a heating pad. An eye cream (Vitamine A, Bausch&Lomb) and a local anesthetic (Xylocain Gel 2%, AstraZeneca) were applied. Atropine (0.3 µg/g of body weight) and dexamethasone (2 µg/g of body weight) were injected subcutaneously. The skull was cleared and a bonding agent (iBOND Total Etch, Heraeus Kulzer) applied. A hook for head fixation was implanted by first applying a droplet of super-glue (Ultra Gel, Pattex). The hook was fixated using light curable dental cement (Tetric EvoFlow, ivoclar vivadent). Betadine was applied to the wound. Antibiotics (100 µg/g of body weight, ceftriaxone, Rocephin, Roche) and pain killers (5 µg/g of body weight, Metacam, Boehringer Ingelheim) were injected subcutaneously before animals woke up. In the second surgery, the skull was thinned above the right monocular visual cortex, determined by stereotaxic coordinates. The eye cream was carefully removed and optical intrinsic imaging was performed to map V1 (see below). After making a craniotomy (3 or 4 mm diameter), 2–3 injections of 150 nl of AAV2/1-*hsyn*-GCaMP6m were made based on the intrinsic imaging and a glass coverslip was positioned. Each experiment consisted of an awake followed by an anesthetized part. For the latter we used isoflurane (0.4–1%). Throughout the anesthetized part, body temperature was measured and maintained at 38°C with a heating pad.

For the two-photon experiments on mice engaged in visually-guided behavior (grating-irrelevant and grating-relevant condition), surgical procedures have been described elsewhere (*Leinweber et al., 2014*). Briefly, the animals were anesthetized with an i.p. injection of a mixture of Fentanyl (Sublimaze, 0.05 µg/g of body weight), Midazolam (Dormicum, 5.0 µg/g of body weight) and Medetomidim (Domitor, 0.5 µg/g of body weight). A craniotomy was made over the right monocular V1, determined by stereotaxic coordinates. The mice were injected with 5 injections of 100–200 nl of AAV2/1-*ef1α*-GCaMP6f, before the coverslip was positioned. Finally, a head plate was implanted.

## Electrophysiological recordings and optogenetic stimulation

Extracellular recordings were made using a multi-tetrode array (Neuronexus, Michigan, A4 × 2-tet-5mm-150-200-121) that was perpendicularly inserted into the brain with a computer-controlled micromanipulator (Scientifica, UK). The probe consisted of 4 evenly spaced shanks, spanning 600 µm of visual cortex in a medial-lateral plane. Each shank contained eight electrode sites, split between two tetrode configurations that were separated by a vertical distance of 150 µm. A reference electrode was also inserted into the cortex, away from the recording site, via a separate craniotomy. In order to target superficial cortical layers, the array was slowly lowered until visually responsive neurons were first encountered. Visual responsiveness was assessed online from multi-unit PSTHs obtained during full-field flash stimuli. Signals were digitized at a sampling frequency of 25 kHz (Tucker Davis Technologies, Florida, RZ2 Bioamp processor). For the optogenetic stimulation of the PV+ cells, illumination (470 nm) was provided by a high-power LED light source (Thorlabs, New Jersey), and directed via a fiber optic cable (400 µm, Thorlabs) which was positioned 3–4 mm from the surface of the cortex, where it dispersed to cover an area approximately 3 mm in diameter. LED illumination was kept constant except for the last 500 ms, where the intensity instantaneously reduced to 50% and then linearly decreased to zero to avoid rebound activation (*Chuong et al., 2014*).

## Intrinsic signal optical imaging

For the experiments on mice not engaged in visually-guided behavior, optical imaging of intrinsic signals was performed before the virus injection of the calcium indicator. Anesthetized mice were placed in front of a monitor and the cortical surface was illuminated with a 630 nm LED light (Thorlabs). The angle of the monitor was ~45° with respect to the craniocaudal axis of the mice with a distance of 20 cm between the center of the screen and the left eye of the mice. The position of the monitor with respect to the mice was kept constant in the following two-photon experiments. In a circular region with a diameter of 10° in the center of the monitor, a square-wave grating was presented for 5 s. Reflectance images were collected through a 4x objective (Olympus, Japan) with a CCD camera (Toshiba, Japan, TELI CS3960DCL). Intrinsic signal changes were analyzed as fractional reflectance changes relative to the prestimulus average. Injections of the calcium indicator were made based on the intrinsic signals.

## Two-photon calcium imaging

For the experiments on mice not engaged in visually-guided behavior, fluorescence was measured with a custom-built two-photon microscope controlled by HelioScan (www.helioscan.org) (*Langer et al., 2013*). The scanhead was based on an 8 kHz resonant scanner (Cambridge Technology, Switzerland), used in bidirectional mode. Images were acquired at 77.7 Hz with a resolution of 200 by 200 pixels. The illumination light source was a Ti:sapphire laser (MaiTai HP, Newport Spectra Physics, California). The excitation wavelength was set to 940 nm or 960 nm. Laser power under the objective (Nikon, Japan, 16 × 0.8 NA) never exceeded 50 mW (laser pulse width ≤100 fs at a repetition rate of 80 MHz). A volume stack was acquired at every imaging site.

For the two-photon experiments on mice engaged in visually-guided behavior (grating-irrelevant and grating-relevant condition), fluorescence was measured with a custom-built two-photon microscope (https://sourceforge.net/projects/iris-scanning/) (*Leinweber et al., 2014*). The scanhead was based on an 8 kHz resonant scanner (Cambridge Technology), used in bidirectional mode. This enabled frame rates of 40 Hz at 400 by 600 pixels. A high-power objective z-piezo stage (Physik Instrumente, Germany) was used to move the objective down in steps of approximately 20 μm between frames and return to the initial position after four frames. With this system, we acquired data at four different depths, reducing the effective frame rate from 40 Hz to 10 Hz. Data were acquired with a 250 MHz digitizer (National Instruments, Texas) and pre-processed with a custom programmed (https://sourceforge.net/projects/iris-scanning/) FPGA (National Instruments). The illumination light source was a Ti:sapphire laser with a prechirp unit (MaiTai eHP DS, Newport Spectra Physics). The excitation wavelength was set to 910 nm. Laser power under the objective (Nikon 16 × 0.8 NA) never exceeded 50 mW (pulse width ≤70 fs at a repetition rate of 80 MHz).

## Treadmill, eye-tracking and visual stimulation

For the electrophysiology experiments, visual stimuli were generated using the open-source MATLAB (MathWorks) Psychophysics Toolbox (*Brainard, 1997*). Drifting square-wave gratings (3 Hz, 0.04 cpd, 100% contrast) moving in eight different directions were presented on an LCD monitor (isoluminant at 82 cd/m²).

Then, responses to stimulus blocks of 7 s were measured. Stimulus blocks were interspersed with 3 s of grey screen. Baseline values were obtained from the 2 s time window before each stimulus. On alternate trials, cortex was optogenetically silenced during the first 3.5 s (see above). The stimuli were presented 5–40 times each.

For the two-photon calcium imaging experiments on mice not engaged in visually-guided behavior, head-restrained mice were placed on a spherical air-supported treadmill (*Dombeck et al., 2007*), which allowed the mice to run or rest at their whim. Visual stimuli were generated using the open-source MATLAB (MathWorks) toolbox StimServer (*Muir and Kampa, 2014*). Drifting sinusoidal gratings (1.5 Hz, 0.04 cpd, 80% contrast) moving in eight different directions were presented (2 s grating interleaved with 4 s grey screen) on a LED-backlit monitor (BenQ XL2410T, iso-luminant at 23 cd/m²). The power-source of the LED-backlight was synchronized with the resonant scanner turn-around points (when data are not acquired) to minimize light-leak from the monitor (*Leinweber et al., 2014*). An iso- and cross-orientation (with an angular difference of 90°) were chosen for the adaptation paradigm. We presented a grating for 10 s at the iso- or cross-orientation at

50% contrast followed by a grating for 10 s at the iso-orientation at 25% or 100% contrast. This resulted in a total of 4 stimulus conditions which were presented in a pseudo-random order for 13–31 times each. The stimulus conditions were interleaved with an iso-luminant grey screen for at least 10 s. Subsequently, the orientation and contrast adaptation paradigms were repeated under anesthesia (presented 30–48 times each). Throughout all imaging sessions, we measured running speed and eye movements. Saccades were detected using a CMOS based video camera at 30 Hz (Imaging Source, North Carolina, DMK 22BUC03). Pupil position was computed offline by smoothing and thresholding the images and fitting a circle to the pupil. The filter radius and the image threshold were adapted manually for each experiment. Pupil position was filtered using a median filter. Eye movements were detected automatically by applying an adapted threshold. This method was cross-validated in several experiments using manual detection of eye movements.

For the experiments on mice engaged in visually-guided behavior with feedback coupling (grating-irrelevant condition), we first mapped the toroidal screen onto the cortical surface using intrinsic optical signal imaging. Single horizontal and vertical bars were shown moving over the whole surface of the screen. The treadmill, eye-tracking and visual stimulation have been described previously (*Leinweber et al., 2014*; *Dombeck et al., 2007*). Briefly, head-restrained mice ran on a spherical air-supported treadmill. Throughout all imaging sessions, we measured the trajectories of the mice in the tunnel and eye movements with a CMOS based video camera at 30 Hz (Imaging Source, DMK 22BUC03). Mice were learning to use a part of the visual field to navigate to a target location in a virtual reality environment. Each mouse had five training sessions on consecutive days (sessions were spaced by 16–32 hr). Starting two days before the first experimental session, mice were water restricted and given a total of at least 1 ml water daily. The weight of the mice was measured daily before and after the training sessions. Before each session, orientation tuning was measured (4 s grating interleaved with 4 s grey screen). During the training session, movement in the virtual tunnel was coupled to the movements of the mice on the spherical treadmill. Mice were trained to orient and run to the end of the tunnel for a water reward (~10 µl per reward) and were immediately teleported back to the start after passing the end of the tunnel. The difficulty of the task (length of the tunnel) was increased during learning to keep the number of rewards approximately constant (~100 per session). Fraction of time spent running across sessions was kept approximately stable by applying occasional air-puffs. Throughout all imaging sessions, we presented a horizontal sinusoidal moving grating (both directions) at 100% contrast in a circular patch (50 degrees in diameter) centered on the retinotopic location of the recording site (45 degrees to the left from the point of view of the mice). This probe patch took up only about an eighth of the entire field of view of the toroidal screen (approximately 200 degrees horizontally, 90 degrees vertically). Grating presentations in the probe patch lasted 10 s (120–163 repetitions per session) and were interspersed with random intervals of grey (10–20 s). Presentations of the drifting grating were not coupled to the behavior of the mice.

For the grating-relevant condition, we repeated the experiment in a new set of mice with two differences. First, the movement in the tunnel was not coupled to their movement on the treadmill but was an exact replay of the visual stimulation used for a mouse under grating-irrelevant condition. The six mice under grating-relevant condition were matched to 6 of the 7 mice under grating-irrelevant condition. Second, the mice were not rewarded at the end of the virtual tunnel but 1 s after the offset of the grating.

The experimental paradigm was chosen to allow us to direct the attention of the mouse either away from or towards the gratings stimulus. One potential concern with a choice of paradigm in which the animal has control of the visual flow feedback in the grating-irrelevant condition is that the difference between predicted and actual visual feedback in the probe patch could result in a mismatch response (*Keller et al., 2012*). Mismatch responses are confined to spatially localized regions in visual space that align to the visual retinotopy (*Zmarz and Keller, 2016*). For this reason, grating-relevant and grating-irrelevant conditions were designed to have an equivalent visual flow mismatch in the retinotopic region of the probe patch. Therefore, any potential influence of mismatch responses is equivalent in both conditions. Moreover, neurons are either mismatch responsive, visually driven, or driven by a combination of both (*Zmarz and Keller, 2016*). The neurons we select for in our analysis are the most visually responsive neurons and hence are unlikely to respond to mismatch (*Zmarz and Keller, 2016*).

## Analysis of electrophysiological data

Electrophysiological data were processed using Matlab (Math-Works) using custom-written code. Single unit spikes were isolated. To this end, the channels were bandpass filtered between 500 Hz and 5000 Hz and tetrodes were whitened. We identified potential spikes using an action potential detector described elsewhere (*Choi et al., 2006*). Then, we performed a principal component analysis (PCA) for each channel using the open-source cluster analysis program KlustaKwik (http://klusta-team.github.io/klustakwik/) (*Watters and Reeke, 2014*). Clusters of potential spikes were determined based on the first three components of the PCA. We calculated the isolation distance of each cluster (*Schmitzer-Torbert et al., 2005*) and excluded clusters with an isolation distance below 20. The number of potential spikes in the poorly isolated multi-unit activity for each tetrode was always at least as large as the number of spikes in any single-unit cluster. Spike times were determined with a 1 ms resolution.

The preferred stimuli and cell types were determined using the average responses over the first 3.5 s of visual stimulation (see Treadmill, eye-tracking and visual stimulation). For each neuron (total 210 cells), we determined the preferred cardinal orientation. Cells were excluded if they failed to respond in at least half of the trials of their preferred cardinal orientation. Then, we compared the average responses to their preferred cardinal orientation in the presence and absence of the optogenetic stimulation. Cells that had a higher average response during optogenetic stimulation were classified as PV+ cells and putative excitatory cells otherwise (data not shown).

All traces of spike rates were binned (~333 ms). Slopes of adaptation (*Figure 1—figure supplement 2a,b*) were estimated by performing a linear regression over 7 s of visual stimulation after normalizing.

## Analysis of two-photon calcium imaging data

Two-photon calcium images were processed using custom written MATLAB (Math-Works) software.

For the experiments on mice not engaged in visually-guided behavior, we used the open-source toolbox FocusStack (https://bitbucket.org/DylanMuir/twophotonanalysis/) (*Muir and Kampa, 2014*). Cells were manually selected using ImageJ (National Institute of Mental Health, NIH). All traces were filtered using a sliding block filter (20 data points corresponding to ~0.26 s). Fluorescence changes ($\Delta F/F$) were calculated as $(F-F_0)/F_0$ using a 2 s baseline before the stimulus to determine $F_0$. To determine orientation tuning curves, responses were calculated as averages over the whole 2 s presentation of visual stimulus. Preferred orientations were determined by fitting a sum-of-Gaussians to single-cell tuning curves. The Gaussians were forced to peak 180° apart and to have the same tuning width. Cells were classified as tuned to the iso- or cross-orientation (see Treadmill, eye-tracking and visual stimulation) depending on which was closer to the peak of the Gaussian fit. Cells were classified as responsive if in at least 50% of the trials the responses to the preferred orientation (iso- or cross-orientation) were significantly above baseline (Z score >2.58 corresponding to p<0.01). Fluorescence changes ($\Delta F/F$) for contrast tuning were calculated using a 3 s baseline before the stimulus.

Slopes of adaptation of tuned cells in *Figure 1—figure supplement 2c,d* were estimated by performing a linear regression over 1–7 s of visual stimulation after normalizing. The initial rise (approximated by 1 s) was excluded from the fit. Adaptation in awake and anesthetized mice (*Figure 1* and *Figure 1—figure supplement 3*) was compared using neurons tuned in both states. Slopes of adaptation in *Figure 1d* were estimated by performing a linear regression over 1–9.75 s of visual stimulation (9.75–10 s was excluded due to filtering). Trials were classified as 'running' if at least during half the visual stimulation the running speed of the mice exceeded 1 cm/s and 'resting' otherwise. Trials were classified as 'eye movement' trials if the mice made at least one saccade during the visual stimulation and 'eye movement-free' otherwise. Cross- and iso-orientation adaptation in anesthetized and awake mice were compared using neurons that were tuned in anesthetized or awake mice, respectively (in *Figure 1—figure supplement 2e–h* and *Figure 1—figure supplement 2i–l*, respectively).

For the experiments on mice engaged in visually-guided behavior (grating-irrelevant and grating-relevant condition), analysis of functional imaging data was conducted as described previously (*Keller et al., 2012*). Briefly, data were full-frame registered using a custom written software (https://sourceforge.net/projects/iris-scanning/). Cells were selected manually based on mean and maximum projections. Raw fluorescence traces were calculated as the average fluorescence of all

pixels within a selected region for each frame. To calculate the fluorescence changes (ΔF/F), the 8-percentile value of the fluorescence distribution in a ±15 s window was subtracted from the raw fluorescence signal, which was then divided by the median of each cell's fluorescence distribution (*Dombeck et al., 2007*). Responses were calculated as averages over the whole 4 s presentation time of visual stimulus. Preferred orientations were determined by fitting a sum-of-Gaussians to single-cell tuning curves averaged over all sessions. Gaussians were fixed to peak 180° apart and to have the same tuning widths. Cells were classified as tuned to the horizontal grating if the peak of the Gaussian fit was within horizontal ±45°. For all activity traces in *Figure 2* and *Figure 2—figure supplement 2*, average activity during a pre-stimulus baseline of 2 s was subtracted. Cells were classified as responsive if in at least half of the sessions the average responses to the horizontal grating were significantly above or below baseline ($|Z$ score$| > 3.29$ corresponding to $p<0.001$). To match the initial conditions of the grating-relevant and grating-irrelevant conditions, 10% of the neurons in the grating-relevant, reward anticipating and non-anticipating condition were excluded. We excluded cells which in session one showed the largest deviations from the mean response in the grating-irrelevant condition. Note that this did not change the results. The performance was quantified as the fraction of time spent running (>1 cm/s) in the direction of the goal with a tolerance of ±25°. Slopes of adaptation and mean responses were estimated by performing a linear regression and averaging over 1–10 s of visual stimulation, respectively (*Figure 2g,h* and *Figure 2—figure supplements 1* and *2g,h*). To estimate the slopes and means, trial responses were divided into four bins per session. Exponential fits were done based on the binned data (four bins per session).

All lick frequencies were baseline-corrected by subtracting the mean lick frequency 15 s to 13 s before the reward. The pre-reward licking (*Figure 2—figure supplement 2a–c*) was defined as the baseline-corrected lick frequency 0.5 s to 0 s before the reward.

## Acknowledgements

This work was supported by EU SECO Grant EU216593, ETH Grant 2-73246-8, and SNF Sinergia to Kevan AC Martin, the FP7 EU Grant 269921 ('BrainScaleS') (FH and BMK), the Novartis Research Foundation (GBK), the Swiss National Science Foundation (GBK), the European Research Council Grant ('NeuroVision', 616509, TDM-F), the Wellcome Trust Grant 095074 (TDM-F), and the Wellcome Trust Research Fellowship WT086697MA (NAL). We thank Kevan AC Martin for his inestimable support in discussions, funding, and resources. We thank Morgane MF Roth for critical comments on the manuscript.

## Additional information

### Competing interests

TDM-F: Reviewing editor, *eLife*. The other authors declare that no competing interests exist.

### Funding

| Funder | Grant reference number | Author |
| --- | --- | --- |
| Seventh Framework Programme | Grant 269921 | Björn M Kampa<br>Fritjof Helmchen |
| Wellcome Trust | Research Fellowship WT086697MA | Nicholas A Lesica |
| European Research Council | Grant 616509 | Thomas D Mrsic-Flogel |
| Wellcome Trust | Grant 095074 | Thomas D Mrsic-Flogel |
| Novartis Foundation | | Georg B Keller |

The funders had no role in study design, data collection and interpretation, or the decision to submit the work for publication.

### Author contributions

AJK, RH, Conceptualization, Data curation, Software, Formal analysis, Validation, Investigation, Visualization, Methodology, Writing—original draft, Writing—review and editing; BMK, NAL, TDM-F,

GBK, FH, Conceptualization, Resources, Data curation, Software, Supervision, Funding acquisition, Validation, Visualization, Methodology, Writing—original draft, Writing—review and editing

## Author ORCIDs

Andreas J Keller, http://orcid.org/0000-0001-7997-6118
Björn M Kampa, http://orcid.org/0000-0002-4343-2634
Nicholas A Lesica, http://orcid.org/0000-0001-5238-4462
Georg B Keller, http://orcid.org/0000-0002-1401-0117
Fritjof Helmchen, http://orcid.org/0000-0002-8867-9569

## Ethics

Animal experimentation: All experiments and surgical procedures were carried out in accordance with the UK Animal (Scientific Procedures) Act under project license 70/7573, approved by the Cantonal Veterinary Office of Zurich, Switzerland, under license number 62/2011, or by the Cantonal Veterinary Office of Basel-Stadt, Switzerland, under license number 2537.

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
