## [Decision Letter]

[Editors’ note: this article was originally rejected after discussions between the reviewers, but the authors were invited to resubmit after an appeal against the decision.]

Thank you for submitting your work entitled "Stimulus relevance modulates contrast adaptation in visual cortex" for consideration by *eLife*. Your article has been reviewed by two peer reviewers, individuals that would agree are leading experimentalists, and the evaluation has been overseen by a Reviewing Editor and a Senior Editor.

Our decision has been reached after consultation between the reviewers. Based on these discussions and the individual reviews below, we regret to inform you that your work will not be considered further for publication in *eLife*.

While the reviewers find considerable merit in your work, they each have voiced strong reservations about the nature of the adaptation as discussed in the detailed comments below. The reviewers further suggest a more fleshed-out study of the mismatch response in the optical flow, as well as a more scholarly presentation of literature results.

*Reviewer #1:*

In their study, Keller et al. address the phenomenon of contrast adaptation in the primary visual cortex in three different brain states, and find evidence that adaptation depends on brain state and/or stimulus relevance. Overall, the study addresses an interesting and important question: might adaptation act as a mechanism to dynamically modulate the gain of cortical responses to relevant vs irrelevant stimuli? They conclude that adaptation might work in this way. The authors primarily rely on two photon calcium imaging from unidentified L2/3 cells in the anesthetized, awake/naïve, and awake/trained mice. Their first result is a striking difference in the contrast adaptation between awake and anesthetized conditions, finding much weaker adaptation in awake/naïve mice. Their second, perhaps more interesting result, is that adaptation in awake conditions reappears for stimuli that become 'irrelevant' over a training task, but then can disappear when that same stimulus becomes 'relevant'. The behavioral paradigm the authors use to make this conclusion is a bit problematic. To measure responses for 'irrelevant' stimuli a grating is presented over the imaged area's receptive field, while the rest of the visual field is a feedback controlled virtual reality environment. I understand why the authors set it up this way, but this necessarily leads to an optic flow mismatch for the receptive field portion. Since prior experiments have shown that optic flow mismatch can strongly modulate V1 responses in mouse, I wonder if this contributed to the effects observed here. To then test adaptation for 'relevant' stimuli, some mice are trained on a movie generated from the 'irrelevant' visual data (from a different mouse) and given a water reward immediately after the previously 'irrelevant' grating appears, presumably providing some 'relevance' as it should act as a classical conditioning cue – which seems to be supported by the anticipatory licking. However, while a clever design, the interpretation is a bit muddied again by the complete mismatch between the virtual visual data and any movement (or lack of movement) of the mouse. It seems like a far simpler and more conventional behavioral paradigm in which a grating is either predictive of reward or not would be much easier to interpret, than a virtual reality system that engages optic flow and locomotion related circuits. Thus, I find the data interesting, but the confusing nature of the behavioral paradigm makes the conclusions the authors propose to draw not entirely convincing.

The authors suggest (mostly tangentially) that PV neurons, which adapt less to gratings, might mediate adaptation. If this were true, then PV neuron activity should also strongly depend on the relevance of the grating stimulus during the behavioral paradigm. Was this true? Perhaps the authors do not have this data, but labeling the PV cells with a reporter would make this trivial, and substantially add to any proposed mechanism for adaptation. Alternatively, these experiments could just as easily be done with extracellular probes and analysis of FS units. Of course, the synapse between PV neurons and excitatory neurons is known to show strong adaptation and changes in synaptic adaptation could well be important as well, even if adaptation at the level of spiking does not correlate well with stimulus relevance.

*Reviewer #2:*

This is a very impressive paper that is aimed at understanding visual adaptation to grating stimuli in the mouse primary visual cortex. The use of optogenetic methods in this paper is very powerful. However, there are significant questions about the interpretation of the data.

The main question is whether or not adaptation could have been taking place in the awake mice and in the mice in which the grating stimuli were task relevant. A secondary question is whether the adaptation phenomena measured mouse cortex are relevant to adaptation in primates including humans.

The stimuli used to test for adaptation were gratings of high contrast. Possibly the responses of the awake animals were hitting a response ceiling, masking adaptation that was taking place but was not apparent because of the response ceiling. Figure 1—figure supplement 2 offers some evidence that a response ceiling is a possibility. In that figure, if one compares the response to 50% contrast in the first 10s with response to 25% contrast in the second 10s following ortho presentation of 50%, one sees they are almost the same. The authors need to convince us that there was no response ceiling, with other experiments.

The question of whether the results can be generalized to humans is based on the history of experiments on visual pattern adaptation. The very first studies of pattern adaptation by Blakemore and Campbell (and many subsequent psychophysical studies) were done in awake humans who were adapted to patterns that were task relevant. In those experiments, there was adaptation not facilitation to exposure to high contrast patterns before behavioral tests. It is not clear how the present results could explain pattern adaptation in those old experiments. Adaptation is also used as a tool frequently in fMRI experiments on awake humans, and again it is not clear how the present results in mouse V1 are consistent with those many adaptation experiments in awake humans. The authors should address this question and explain what they think the relevance of their results is to prior results on human adaptation.

In addition, there is another more detailed question about Figure 1—figure supplement 2. It is not clear to me why the measure of adaptation is larger for the Exc-cell 2-photon data in Figure 1—figure supplement 2 than for the electrophysiological data in Figure 1—figure supplement 2. In A the adaptation seems clearly greater than in C, yet the bar graphs in B and D state the opposite. Are the vertical scales in error in B and D?

---

## [Author Response]

[Editors’ note: the author responses to the first round of peer review follow.]

*Reviewer #1:*

*In their study, Keller et al. address the phenomenon of contrast adaptation in the primary visual cortex in three different brain states, and find evidence that adaptation depends on brain state and/or stimulus relevance. Overall, the study addresses an interesting and important question: might adaptation act as a mechanism to dynamically modulate the gain of cortical responses to relevant vs irrelevant stimuli? They conclude that adaptation might work in this way. The authors primarily rely on two photon calcium imaging from unidentified L2/3 cells in the anesthetized, awake/naïve, and awake/trained mice. Their first result is a striking difference in the contrast adaptation between awake and anesthetized conditions, finding much weaker adaptation in awake/naïve mice. Their second, perhaps more interesting result, is that adaptation in awake conditions reappears for stimuli that become 'irrelevant' over a training task, but then can disappear when that same stimulus becomes 'relevant'. The behavioral paradigm the authors use to make this conclusion is a bit problematic. To measure responses for 'irrelevant' stimuli a grating is presented over the imaged area's receptive field, while the rest of the visual field is a feedback controlled virtual reality environment. I understand why the authors set it up this way, but this necessarily leads to an optic flow mismatch for the receptive field portion. Since prior experiments have shown that optic flow mismatch can strongly modulate V1 responses in mouse, I wonder if this contributed to the effects observed here.*

We thank the reviewer for the assessment that we ‘address an interesting and important question’. The point about a potential effect of the mismatching visual input is important and very interesting. However, we do not think that visual flow mismatch is confounding the interpretation of our results for the following reasons:

1) Mismatch responses are confined to spatially localized regions in visual space that align to the visual retinotopy (Zmarz and Keller, 2016, note, this paper was not published at the time of first submission). In our paradigm, the visual flow mismatch is equivalent (in the retinotopic region of the grating patch) for both “grating irrelevant” (feedback) and “grating relevant” (replay) conditions. Taking these two things together suggests that only local mismatch (which is identical in both conditions) would influence mismatch responses in the retinotopic region of our probe patch. In other words, if mismatch responses would influence activity, their influence should be identical in replay and feedback sessions. However, we observe a strong difference in adaptation in the two conditions (see Figure 2).

2) Neurons are either mismatch responsive, visually driven, or driven by a combination of both (Zmarz and Keller, 2016). The neurons we select for in our analysis, are the most visually responsive neurons in our dataset (based on their responses during passive viewing of orientated gratings) and hence are unlikely to respond to mismatch.

3) Lastly, a key point of our experiments was to show that cortical adaptation is not adequately explained by adaptation of feedforward input. We indeed cannot identify the source or nature of the attentional signal that modulates adaptation in V1, but we can show that it is not purely feed-forward driven (because visual input in playback and feedback conditions is identical).

In summary, the mismatch responses as described by Zmarz and Keller, 2016, or Keller et al., 2012, cannot explain the differences between relevant and irrelevant conditions observed here.

On a side-note: any stimulus that is not coupled to movement generates a “mismatch” (e.g., visual experiments where neural responses are probed with moving stimuli that the animal does not self-generate). The mismatch responses described for example by Zmarz and Keller, 2016, are merely those where there is less visual flow than expected (an experiment on a passively observing animal probes mismatches of the form of more visual flow than expected). Hence, it is likely that a large part of visual processing in general is concerned mainly with detecting and processing mismatch (any visual input that is not the direct consequence of self-motion).

To address this concern, we have added a discussion about mismatch responses to the description of the grating-relevant and grating-irrelevant conditions in the Materials and methods section (in the section ‘Treadmill, eye-tracking and visual stimulation’, last paragraph).

*To then test adaptation for 'relevant' stimuli, some mice are trained on a movie generated from the 'irrelevant' visual data (from a different mouse) and given a water reward immediately after the previously 'irrelevant' grating appears, presumably providing some 'relevance' as it should act as a classical conditioning cue – which seems to be supported by the anticipatory licking. However, while a clever design, the interpretation is a bit muddied again by the complete mismatch between the virtual visual data and any movement (or lack of movement) of the mouse.*

Please see arguments above for the influence of mismatch.

As for similarity of movement, average fraction of time spent running and running speed were not different in the grating-relevant and grating-irrelevant condition (see Figure 2—figure supplement 2).

Just to avoid a potential misunderstanding, the reward stimulus was not given immediately after the previous ‘irrelevant’ grating appears, but after the offset of the grating with a delay of one second. This was done to separate the grating presentation form the reward and avoid any contamination of the grating responses with lick related signals.

To address the concern of mismatch influencing our results, we have added a discussion to the task description in the Materials and methods section (‘Treadmill, eye-tracking and visual stimulation’, last paragraph).

*It seems like a far simpler and more conventional behavioral paradigm in which a grating is either predictive of reward or not would be much easier to interpret, than a virtual reality system that engages optic flow and locomotion related circuits. Thus, I find the data interesting, but the confusing nature of the behavioral paradigm makes the conclusions the authors propose to draw not entirely convincing.*

This is an interesting suggestion. The reason why we did not pursue this type of experiment is that it would remain unclear if the mouse is indeed less attentive to a grating when simply shown without an associated reward. Hence, we aimed to design an experimental paradigm that would allow us to direct the attention either away from or towards the grating. The virtual navigation task is simply one way to ensure that the animal directs its attention towards the part of the visual field informative for the task.

We do understand the hesitation of the reviewer, but given that, when designing the experiments, we took into consideration that mismatch processing is local (at least to the same extent that visual processing is local, see Zmarz and Keller, 2016). We do not think that non-local mismatch signals are more of a confound than non-local visual signals.

In the Materials and methods section, we now explain the choice of experimental paradigm more thoroughly (‘Treadmill, eye-tracking and visual stimulation’, last paragraph).

*The authors suggest (mostly tangentially) that PV neurons, which adapt less to gratings, might mediate adaptation. If this were true, then PV neuron activity should also strongly depend on the relevance of the grating stimulus during the behavioral paradigm. Was this true? Perhaps the authors do not have this data, but labeling the PV cells with a reporter would make this trivial, and substantially add to any proposed mechanism for adaptation. Alternatively, these experiments could just as easily be done with extracellular probes and analysis of FS units. Of course, the synapse between PV neurons and excitatory neurons is known to show strong adaptation and changes in synaptic adaptation could well be important as well, even if adaptation at the level of spiking does not correlate well with stimulus relevance.*

Please excuse, we were unclear in our phrasing. We intended to show that in the anesthetized mouse, consistent with results in the cat (Keller and Martin, 2015), adaptation could in principle be explained by PV interneurons because these adapt less to gratings. We did not mean to suggest that this is the same mechanism responsible for the behavioral modulation of adaptation. On the other hand, considering changes in adaptation of inhibitory interneurons as potential mechanism as suggested by the reviewer is interesting. In fact, we have a dataset from experiments in VGAT animals (where we have monitored activity of inhibitory neurons) to argue the behavioral modulation of adaptation cannot easily be explained by changes in inhibitory adaptation – hence the source of this signal is unlikely to be local. In these experiments, we find that inhibitory neurons essentially follow the activity of putative excitatory cells. If the reviewer thinks this would be useful, we would be happy to include these data in the manuscript, but think that this would go beyond the scope of the paper.

We now emphasize this more clearly in the Results and discussion section (third paragraph, last sentence) and the figure legend of Figure 1—figure supplement 4.

*Reviewer #2:*

*This is a very impressive paper that is aimed at understanding visual adaptation to grating stimuli in the mouse primary visual cortex. The use of optogenetic methods in this paper is very powerful. However, there are significant questions about the interpretation of the data.*

*The main question is whether or not adaptation could have been taking place in the awake mice and in the mice in which the grating stimuli were task relevant. A secondary question is whether the adaptation phenomena measured mouse cortex are relevant to adaptation in primates including humans.*

*The stimuli used to test for adaptation were gratings of high contrast. Possibly the responses of the awake animals were hitting a response ceiling, masking adaptation that was taking place but was not apparent because of the response ceiling. Figure 1—figure supplement 2 offers some evidence that a response ceiling is a possibility. In that figure, if one compares the response to 50% contrast in the first 10s with response to 25% contrast in the second 10s following ortho presentation of 50%, one sees they are almost the same. The authors need to convince us that there was no response ceiling, with other experiments.*

This is a very good point. Note, however, that the data shown in Figure 1—figure supplement 2 were recorded in anesthetized animals, where we do find adaptation, especially in response to gratings of higher contrasts. To control for ceiling effects, we performed experiments in which we followed the presentation of the 50% contrast grating with a 100% contrast grating. In awake mice, cells do not adapt to the 50% contrast grating, but can further increase their activity if the contrast is increased to 100%. This demonstrates that the lack of adaptation is not due to response ceiling.

To address the concern of a response ceiling, we have added data from these experiments as Figure 1—figure supplement 2 to the manuscript.

*The question of whether the results can be generalized to humans is based on the history of experiments on visual pattern adaptation. The very first studies of pattern adaptation by Blakemore and Campbell (and many subsequent psychophysical studies) were done in awake humans who were adapted to patterns that were task relevant. In those experiments, there was adaptation not facilitation to exposure to high contrast patterns before behavioral tests. It is not clear how the present results could explain pattern adaptation in those old experiments.*

This is a very good point. Many psychophysical studies indeed suggest that the visual system adapts to sustained visual patterns. Psychophysical measurements, however, reflect the activity of the whole brain, not just primary visual cortex. Using fMRI in humans, pattern adaptation has been shown to be stronger in higher visual areas than in primary visual cortex (Huk and Heeger, 2001). Hence, it is possible that adaptation is still present in some higher visual areas independent of attention.

An alternate explanation for this discrepancy is that attention can overcome adaptation. Using psychophysical measurements, Pestilli and colleagues (Pestelli et al., 2007, PMID: 17685805) show that attention and adaptation have opposite effects.

In the Results and discussion section, we now discuss our results in the context of the human literature (fourth paragraph).

*Adaptation is also used as a tool frequently in fMRI experiments on awake humans, and again it is not clear how the present results in mouse V1 are consistent with those many adaptation experiments in awake humans. The authors should address this question and explain what they think the relevance of their results is to prior results on human adaptation.*

Many fMRI studies on awake humans indeed find adaptation in primary visual cortex. This disparity is likely the result of small but relevant differences in study design.

Gardner et al., 2005, show adaptation in primary visual cortex to a checkerboard pattern. In their study, subjects were performing a detection task to guide their attention away from the checkerboard stimulus. Moreover, the checkerboard was flickered, which induces temporal contrast adaptation. This type of adaptation has been demonstrated to already take place in the retina (Smirnakis et al., 1997, Chander and Chichilnisky, 2001).

Fang et al., 2005, also report pattern adaptation in human primary visual cortex using fMRI. Here, they used Gabor patches which were counter-phase flickered. Again, this type of adaptation is already strongly present in the retina.

Kastner et al., 2003, however, find no contrast adaptation in primary visual cortex despite using a flickered checkerboard stimulus. To assess contrast adaptation, they presented ascending versus descending contrasts. Gardner et al., 2005, argue that this method might not be sensitive enough to capture an adaptation effect.

Engel and Furmanski (2001, PMID: 11356883) show that there is selective adaptation to color contrast. Here, they presented low contrast red-green or dark-light sinusoidal moving gratings before and after exposure to high contrast gratings. Although, they demonstrate that there is color-contrast specific adaptation they do not show that spatial luminance contrast adaptation is present in primary visual cortex.

Huk and Heeger, 2001, find a weak adaptation in V1 when participants were attending two separate moving plaid stimuli. This could be explained by the fact that distributing attention decreases attentional effects (Ito and Gilbert, 1999).

In conclusion, the results of our study are largely consistent with the current human fMRI literature and significantly go beyond the current literature. It would be very interesting to see whether and how attention affects adaptation in human primary visual cortex for stimuli that mainly evoke adaptation in cortex (as opposed to stimuli that are known to result in significant adaptation in the retina or thalamus).

In the Results and discussion section, we now discuss our results in the context of the human literature (fourth paragraph).

In addition, there is another more detailed question about Figure 1—figure supplement 2. It is not clear to me why the measure of adaptation is larger for the Exc-cell 2-photon data in Figure 1—figure supplement 2 than for the electrophysiological data in Figure 1—figure supplement 2. In A the adaptation seems clearly greater than in C, yet the bar graphs in B and D state the opposite. Are the vertical scales in error in B and D?

The numbers indicated are indeed correct. To minimize the influence of initial response transients, and to focus on the slow adaptation occurring over the course of seconds, we estimate adaptation using a linear fit to the data (in a window from 0 s to 7 s).